# Device Exposure and Patient Risk Factors’ Impact on the Healthcare-Associated Infection Rates in PICUs

**DOI:** 10.3390/children9111669

**Published:** 2022-10-31

**Authors:** Elena Fresán-Ruiz, Gemma Pons-Tomás, Juan Carlos de Carlos-Vicente, Amaya Bustinza-Arriortua, María Slocker-Barrio, Sylvia Belda-Hofheinz, Montserrat Nieto-Moro, Sonia María Uriona-Tuma, Laia Pinós-Tella, Elvira Morteruel-Arizcuren, Cristina Schuffelmann, Yolanda Peña-López, Sara Bobillo-Pérez, Iolanda Jordan

**Affiliations:** 1Pediatric Intensive Care Unit, Hospital Sant Joan de Déu, 08950 Barcelona, Spainijordan@hsjdbcn.es (I.J.); 2Immunological and Respiratory Disorders in the Pediatric Critical Patient Research Group, Institut de Recerca Sant Joan de Déu, 08950 Barcelona, Spain; 3Pediatrics Department, Hospital Sant Joan de Déu, 08950 Barcelona, Spain; 4Pediatric Intensive Care Unit, Hospital Son Espases, 07120 Palma de Mallorca, Spain; 5Pediatric Intensive Care Unit, Hospital Gregorio Marañón, 28007 Madrid, Spain; 6Pediatric Intensive Care Unit, Hospital 12 de Octubre, 28041 Madrid, Spain; 7Pediatric Intensive Care Unit, Hospital Niño Jesús, 28009 Madrid, Spain; 8Preventive Medicine and Public Health, ENVIN-HELICS Registry Administration, Hospital Vall d’Hebron, 08035 Barcelona, Spain; 9Pediatric Intensive Care Unit, Hospital de Cruces, 48903 Bilbao, Spain; 10Pediatric Intensive Care Unit, Hospital La Paz, 28046 Madrid, Spain; 11Pediatric Intensive Care Unit, Hospital Materno-Infantil Vall d’Hebron, 08035 Barcelona, Spain; 12Consorcio de Investigación Biomédica en Red de Epidemiología y Salud Pública (CIBERESP), 28029 Madrid, Spain

**Keywords:** PICU, Spain, children, healthcare-associated infections, device-associated infections, CLABSI, VAP, CAUTI, HAI Zero Bundles, HAI risk factors

## Abstract

Healthcare-associated infections related to device use (DA-HAIs) are a serious public health problem since they increase mortality, length of hospital stay and healthcare costs. We performed a multicenter, prospective study analyzing critically ill pediatric patients admitted to 26 Spanish pediatric intensive care units (PICUs) over a 3-month period each year from 2014 to 2019. To make comparisons and evaluate the influence of HAI Zero Bundles (care bundles that intend to reduce the DA-HAI rates to zero) on PICU HAI rates, the analysis was divided into two periods: 2014–2016 and 2017–2019 (once most of the units had incorporated all the Zero Bundles). A total of 11,260 pediatric patients were included. There were 390 episodes of HAIs in 317 patients and the overall rate of HAIs was 6.3 per 1000 patient days. The DA-HAI distribution was: 2.46/1000 CVC days for central-line-associated bloodstream infections (CLABSIs), 5.75/1000 MV days for ventilator-associated pneumonia (VAP) and 3.6/1000 UC days for catheter-associated urinary tract infections (CAUTIs). Comparing the two periods, the HAI rate decreased (*p* = 0.061) as well as HAI episodes (*p* = 0.011). The results demonstrate that exposure to devices constitutes an extrinsic risk factor for acquiring HAIs. The multivariate analysis highlights previous bacterial colonization by multidrug-resistant (MDR) bacteria as the most important extrinsic risk factor for HAIs (OR 20.4; 95%CI 14.3–29.1). In conclusion, HAI Zero Bundles have been shown to decrease HAI rates, and the focus should be on the prompt removal of devices, especially in children with important intrinsic risk factors.

## 1. Introduction

Healthcare-associated infections (HAIs) are a major concern in intensive care units (ICUs) since they increase mortality, length of hospital stay and healthcare costs [1,2]. A quarter of these infections occur in patients admitted to ICUs, even though these units only represent 10% of total hospital beds. The same implications occur in pediatric intensive care units (PICUs), where, in terms of patients’ safety, HAIs are an undesirable and a highly dangerous event [3].

There are several risk factors for developing an infection acquired in the PICU, such as requiring surgery, extracorporeal membrane oxygenation (ECMO), renal replacement therapy or parenteral nutrition, presenting neutropenia, or carrying devices (central line, urinary catheter or endotracheal tube, among others) [2,4,5]. In fact, in pediatric population, the most common HAIs are those associated with medical devices: central-line-associated blood stream infections (CLABSIs), ventilator-associated pneumonia (VAP) and catheter-associated urinary tract infections (CAUTIs) [5].

Because of the progressively complex cases and the increase in pediatric comorbidities, surveillance of the HAI taxes and their characteristics is necessary in order to adapt the HAI prevention measures. The international and national surveillance systems allow us to compare HAI data and to implement new measures to improve our practice regarding HAIs [6,7,8].

A surveillance system for HAIs in Spanish PICUs was established in 2007. It was subsequently consolidated in 2013 as the National Nosocomial Infections Surveillance System (Pediatric-ENVIN) within the HELICS project (Hospitals in Europe Link for Infection Control through Surveillance) [9,10]. The data included in this comprehensive registry provide important information not only on overall HAI rates, but also on antibiotic use, microorganism isolates and resistance profile [11]. It is considered as a benchmark for national and international PICUs.

In order to prevent device-associated HAIs, Spanish Intensive Care Units (ICUs) have implemented care bundles that are called “Zero Bundles”. They are simple sets of evidence-based practices that, when applied collectively, improve the reliability of their delivery and patient outcomes [12]. As they are intended to reduce the rate of device-associated HAIs to zero, they are called: Bacteremia Zero, Pneumonia Zero and Urinary Tract Infection Zero Bundles [13,14]. The second major project aimed to reduce the level of bacterial resistance to zero, and that led to the implantation of the Resistance Zero Bundle.

The novelty of this study lies in the limited information available in the literature on HAIs and risk factors in PICUs, and even more limited is the information regarding the influence of HAI Zero Bundles on the control of these infections. This study offers an insight into the epidemiology of patients admitted in PICUs participating in the multicenter Pediatrics-ENVIN-HELICS registry. Our aim is to compare the HAI rates over time, taking into account the hypothesis that the implementation of the HAI Zero Bundles may have reduced the incidence of HAIs.

## 2. Materials and Methods

A prospective, multicenter observational study was conducted in 26 PICUs in Spain during a 3-month period every year (from 1 April to 30 June, according to ENVIN-HELICS surveillance criteria) from 2014 to 2019. The reason for choosing these particular months is that they comprise a time of the year with an average workload, so the data are considered to be more accurate. Subjects included were hospitalized pediatric patients (≥1 month and ≤18 years of age) who required admission to the PICU during the study period. All patients admitted before or after the study period were excluded.

We included all the patients registered by all PICUs participating in the National Nosocomial Infection Surveillance System. However, at the beginning of the data collection period, not all of these PICUs had yet implemented HAI Zero Bundles. We conducted a recurrent survey among these 31 units to determine exactly when they had implemented HAI Zero Bundles.

The results show that the implementation of the different bundles was inconsistent. By 2016, 90.3% (*n* = 28) of the units had established the Bacteremia Zero Bundle, but only 64.5% (*n* = 20) had deployed the Pneumonia Zero Bundle, 35.5% (*n* = 11) the Urinary Tract Infection Zero Bundle and 48.4% (*n* = 15) the Resistance Zero Bundle. The scenario was different by 2019: 93.5% (*n* = 29) had the Bacteremia Zero Bundle, 80.7% (*n* = 25) the Pneumonia Zero Bundle, 67.7% (*n* = 21) the Urinary Tract Infection Zero Bundle and 74.2% (*n* = 23) the Resistance Zero Bundle.

Thus, to test the hypothesis that HAI Zero Bundles can influence HAI rates, the analysis was split into two 3-year periods, 2014–2016 and 2017–2019, for comparison.

### 2.1. Definitions

Comorbidities: The following underlying diseases were included: diabetes, neoplasia, renal failure, immunosuppression, chronic obstructive pulmonary disease, cirrhosis, malnutrition and transplantation.

Healthcare-associated infections [15,16], HAIs, were divided between those contracted outside the PICU and inside the PICU, according to the site of onset of infection. We expressed the rate of HAI as HAIs/1000 patient days. Patient days, as the name suggests, is the total number of days that the patients spent in the PICU.

A device-related HAI was diagnosed in the patients who carried a device (endotracheal tube, central line or indwelling urinary catheter) that was used in the 48 h period prior to the onset of the infection, even if it was only used intermittently [17,18]. Table 1 lists the different definitions of device-associated infection HAIs registered: primary bloodstream infections (including CLABSI and unknown-source bacteremia (USB)), VAP and CAUTI, using the ENVIN diagnostic criteria adapted to pediatrics based on the Centre of Disease Control recommendations [1,10]. USB is an episode of bacteremia for which it is not possible to identify any cause (catheter or other causes).

HAI risk factors:

Patients were considered positive if they presented one or more of the following risks of infection: receiving antibiotics prior to PICU admission, previous surgery, urgent surgery during PICU stay, mechanical ventilation, urinary catheter, external ventricular shunt, renal replacement therapy, parenteral nutrition, neutropenia or extracorporeal membrane oxygenation (ECMO).

### 2.2. Outcome

The first outcome was the determination of changes in device-associated HAI epidemiology over time, considering the influence of the implementation of HAI Zero Bundles. The second outcome was the detection of which risk factors were associated with HAIs.

The considered device exposure indicators were:-Device days: defined as the duration of use of a device (in days).-Device-associated infection (DAI): defined as infection associated with the use of a device.-Device-associated rate per 1000 days of device exposure: defined as the number of infections associated with the use of a device divided by the duration of use of a device (in days).-Device utilization ratio (DUR): defined as the duration of use of a device (in days) with respect to the global number of days in the PICU [10].-Cumulative incidence: number of infections divided by the number of patients that used the device.

### 2.3. Variables

During the study period, the physicians of the participating hospitals entered all the clinical data for each patient admitted to the PICU into a standardized online registry. Parameters collected for this analysis were: demographic characteristics, diagnosis at admission, pediatric mortality risk score (PRISM), Glasgow score at admission, comorbidities, risk factors for infection, community-acquired infections, HAIs acquired outside and inside the PICU, device-associated HAIs, microorganisms responsible for the infection, antibiotics used, antibiotic susceptibility profile and length of stay (LOS) [10].

### 2.4. Statistical Analysis

The recorded data were mapped on a specific server (https://hws.vhebron.net/envin-helics/ (accessed on 1 April 2014). Categorical variables were reported as frequency (*n*) and percentage (%), while continuous variables were summarized as median and interquartile range (IQR) because they did not have a normal distribution. For bivariate analysis, qualitative variables were compared using the χ^2^ test or Fisher’s exact test; continuous variables were compared using the Mann–Whitney U-test or Student’s t-test when applicable. The correlation coefficient was calculated using Spearman’s rank correlation coefficient. Once computed, the correlation matrix was plotted using a heatmap where blue indicates strong correlations (coefficient close to 1) and red indicates negative correlations (coefficient close to −1). A forward stepwise logistic regression analysis was conducted with the variables that had a *p* value of less than 0.01 in the univariate analysis to detect independent HAI risk factors. Probability values of less than 0.05 were considered statistically significant. The statistical analysis was performed using IBM SPSS 25.0 Statistics^®^.

## 3. Results

### 3.1. Patients and Clinical Characteristics

The total number of patients included in the study was 11,260:5349 (47.5%) in the first period and 5911 (52.5%) in the second period when most of the units had implemented the Zero programs. The median PRISM III was 2 (IQR 0–5). The median age was 43 months (IQR 10–115), and 6368 (56.6%) were males. No differences were detected between age and gender in either group. An underlying disease was recorded in 2317 children (20.6%). The reason for admission varied among years, and only statistically significant differences were detected for urgent surgery during the two periods. Table 2 includes the main clinical characteristics of the sample and compares the two groups as regards these variables. Table 3 describes the healthcare-associated infection risk factors and mortality rate and also compares the two groups.

### 3.2. HAI Risk Factors

During the period of study, the presence of HAI risk factors changed, but when comparing the two periods, a global decrease was observed (86.1% vs. 83.8%, and also for each risk factor). Only a subtle increase in the frequency of antibiotics prior to PICU admission and previous surgery was detected. Details are included in Table 2.

### 3.3. HAI Rates

There were 390 episodes of HAIs (3.46%) during the period of the study in 317 patients. The ratio of patients who developed at least one HAI tended to decrease in the second period (2014–2016: 167, 3.1% vs. 2017–2019: 150, 2.5%, *p* = 0.061), and statistically significant differences were detected in the proportion of HAI episodes between both groups: 210 (3.9%) vs. 180, (3.1%), with *p* = 0.011.

The global HAI rate was 2.46/1000 CVC days for CLABSIs, 5.75/1000 MV days for VAP and 3.6/1000 UC days for CAUTIs. Data regarding the device exposure, device utilization ratio and the specific rate per year are detailed in Table 4.

A trend towards a decrease in the HAI rate was observed until 2019, when a new peak was observed, whereas the device utilization ratio for each device remained stable. Figure 1 shows both HAI rates and device utilization ratio per year.

In comparing the two periods (2014–2016 vs. 2017–2019), a decrease in all the cumulative incidences was observed, this being especially evident for USB: CLABSIs, 0.879 vs. 0.727, VAP 1.103 vs. 0.863, CAUTIs 1.047 vs. 0.829 and USBs 2.206 vs. 0.659.

The global cumulative incidence was 0.799 for CLABSIs, 0.977 for VAP and 0.933 for CAUTIs and 0.755 for USB. The cumulative incidence for each HAI per year showed a steady global decrease as is represented in Figure 2, with the exception of CLABSIs and CAUTIs, which presented a sharp rise in 2019. In comparing the two periods (2014–2016 vs. 2017–2019), a decrease in all the cumulative incidences was observed, this being especially evident for USB: CLABSIs, 0.879 vs. 0.727, VAP 1.103 vs. 0.863, CAUTIs 1.047 vs. 0.829 and USB 2.206 vs. 0.659.

Dividing the sample according to age, the cumulative incidence for each HAI was lower in children older than 2 years of age, as is represented in Figure 3.

### 3.4. Univariate Analysis of Risk Factors for HAI

In the univariate analysis, the proportion of HAI risk factors was statistically significant higher in patients that developed an HAI later. Only previous surgery had a similar distribution in patients who developed an HAI and patients who did not. All these results are included in Table 5.

Additionally, the proportion of previous colonization by multidrug-resistant (MDR) bacteria was statistically significant higher in the HAI group: 26.2% vs. 0.9%, with *p* < 0.001. The most frequent (12.6%) cause of colonization by drug-resistant bacteria were extended-spectrum beta-lactamase-producing bacteria (ESBL). The duration of device exposure was statistically significant longer in patients with HAIs: 18 days (IQR 8–30) vs. 0 (IQR 0–3) for CVC; 11 days (IQR 4–23) vs. 0 (IQR 0–1) for endotracheal tubes; and 11 days (IQR 6–20.5) vs. 0 (IQR 0–3) for urinary catheters, all with *p* < 0.001. Additionally, the length of stay was longer in patients who developed HAIs: 23 days (IQR 13.5–39.5) vs. 3 (IQR 2–6), *p* < 0.001.

### 3.5. Multivariate Analysis of Risk Factors for HAI

In the multivariate analysis (Figure 4), the detected intrinsic risk factors for HAIs were: previous bacterial colonization (OR 20.4; 95%CI 14.3–29.1), need for urgent surgery (OR 3.5; 95% CI 2.6–4.7), medical reason for admission (OR 1.5; 95% CI 1.1–1.9), other PICUs as a referring service (OR 2.1; 95% CI 1.3–3.3), presence of a comorbidity (OR 1.5, 95%CI 1.2–2.0) and higher PRISM III score (OR 1.03, 95% CI 1.02–1.04).

Regarding analysis of extrinsic risk factors for HAIs during PICU stay, Figure 5 represents the different correlations of devices regarding the presence or absence of HAIs.

The correlation coefficient was calculated using Spearman’s rank correlation coefficient. A color scale has been used to represent the correlations: blue for positive correlations and red for negative correlations. CVC: central venous catheter; HAI: healthcare-associated infections; LOS: length of stay; MV: mechanical ventilation; UC: urinary catheter.

## 4. Discussion

This is, as far as we know, one of the few prospective multicenter studies conducted in European PICU patients that describe the epidemiology and risk factors of device-associated HAIs, and the first to analyze the evolution of the incidence of these HAIs after the implementation of HAI Zero Bundles.

Different organizations have made great efforts to prevent and control HAIs worldwide [18,19,20,21]. The available data vary enormously depending on the socioeconomic level of the different countries. As the World Health Organization [22,23,24] describes, very scanty data about HAI rates are available from the vast majority of low- and middle-income countries. However, what we can extract from that data is that HAI rates in those ICUs are at least 2–3 times the rate of high-income countries ICUs (42.7 episodes per 1000 patient days vs. 17.0 episodes per 1000 patient days). The number of device-associated HAIs (DA-HAIs) in our ENVIN-based study was lower than that in adult patients, which contrasts with other similar studies in PICUs [25,26,27,28,29]. In this sense, a study including patients admitted to four Greek PICUs [26] describes an overall rate of HAIs of 18.3 per 1000 patient days, almost three-fold the rate found in our study (6.3 per 1000 patient days).

Many publications emphasize that bundle care auditing makes a significant difference in all DA-HAI rates [30]. Our results manifest a significant decrease in HAI episodes and device-associated HAI rates after the implementation of HAI Zero Bundles [28] that promote better device use policies, such as decreasing the indication of devices and the number of days of exposure. Results show that all these measures also led to a decrease in the HAI risk factors.

There is evidence that exposure to devices increases HAIs, as our article demonstrates, and the serious consequences of DA-HAIs in children are beyond question [21,27]. Our study reveals that patients with HAIs have a statistically significant longer length of stay in the PICU and longer device exposure, as does the study published by De Mello et al. [31], in which each day of ventilator use increased the risk of acquiring an HAI by 16% in a PICU.

We emphasize the importance of an adequate policy for the use of antibiotics following the fulfillment of Antimicrobial Stewardship (AMS) programs. Furthermore, receiving antibiotics during PICU stays stands out as another risk factor for acquiring HAIs in the PICU [32].

In addition, this work focuses on the intrinsic risk factors of pediatric patients, for which there is little published literature.

In the multivariate analysis, the most important independent intrinsic risk factor associated with DA-HAIs was previous colonization by an MDR bacterium (OR 20.4; 95%CI 14.3–29.1). In a recent study by Girona et al. [33], 22.5% of the DA-HAIs diagnosed in a tertiary hospital PICU were caused by MDR pathogens. As shown in our data, the most common MDR microorganism was also ESBL bacteria (36.8%), and the most frequent infection was CLABSI.

A study conducted in children admitted to the PICU in Brazil and Italy described that 93.3% of HAI cases were children affected by comorbidities [34]. This is consistent with our study, where the presence of a comorbidity increased the risk of HAI by 1.5 (OR 1.5; 95%CI 1.2–2.0). This could be explained by the fact that children with chronic pathologies are frequently in contact with healthcare facilities, and they may also have an increased risk of a torpid evolution of diseases, involving greater use of devices. Moreover, comorbidities have been shown to be a risk factor for multidrug-resistant bacteria [33], which can make the treatment of these infections very challenging.

Younger age has been reported to be a risk factor for DA-HAIs in previous studies [2,3,31,34,35]. We found that the median age of the patients who developed an HAI was 37 months younger (45 months vs. 8 months). Moreover, we divided our sample according to age and discovered that the cumulative incidence for each HAI is higher in children under 2 years of age, which could be because this is the period of risk for invasive bacterial disease. In the multivariate analysis, age was not found to be an independent risk factor for HAI.

Difficulties were found when comparing our data regarding the pediatric score of mortality, due to the heterogeneity of scores used in the literature. Hatachi et al. [36] described a mean PIM2 among the patients with DA-HAIs that was almost twice as high as that among the patients without DA-HAIs. The pediatric ENVIN-HELICS register PRISM III score and our study reveal that a higher score is an independent risk factor for HAI (OR 1.03, 95% CI 1.02–1.04). This is probably because a greater severity of the illness entails a high density of invasive therapeutic interventions on admission, which means a higher device utilization ratio and prolonged PICU stay, resulting in a higher infection rate [36,37].

From our results, we emphasize the importance of HAI Zero Bundles in PICUs. The results of our study show that HAI Zero Bundles are related to a significant decrease in HAI episodes and device-associated HAI rates, which can only improve our patients’ results in terms of a shorter length of stay, less use of antibiotics, fewer appearances of new multidrug-resistant bacteria and lower morbidity and mortality, with huge implications for patients, their families and public health.

Although this study was carried out at a national level, not European, we think it had a sufficient number of patients and variables to draw relevant conclusions. However, more studies are needed on clinical outcomes, hospital expenses, indirect costs, length of hospital stay, readmissions and mortality in the pediatric population.

## 5. Conclusions

Our data reveal that the exposure to devices increases HAIs, and that patients with device-associated HAIs have statistically significant longer device exposure and also a longer length of stay in the PICU.

This study highlights the importance that implementation of HAI Zero Bundles has, as these programs lead to a decrease in HAI risk factors, HAI episodes and device-associated HAI rates.

The analysis conducted on the intrinsic risk factors that result in higher HAI rates is remarkable. These data could be used to develop HAI risk scores that help stratify patients and provide more personalized medicine. Even more rigorous efforts should be made in terms of device removal and antibiotic therapy optimization with patients that are at higher risk of acquiring an HAI.

## Figures and Tables

**Figure 1 children-09-01669-f001:**
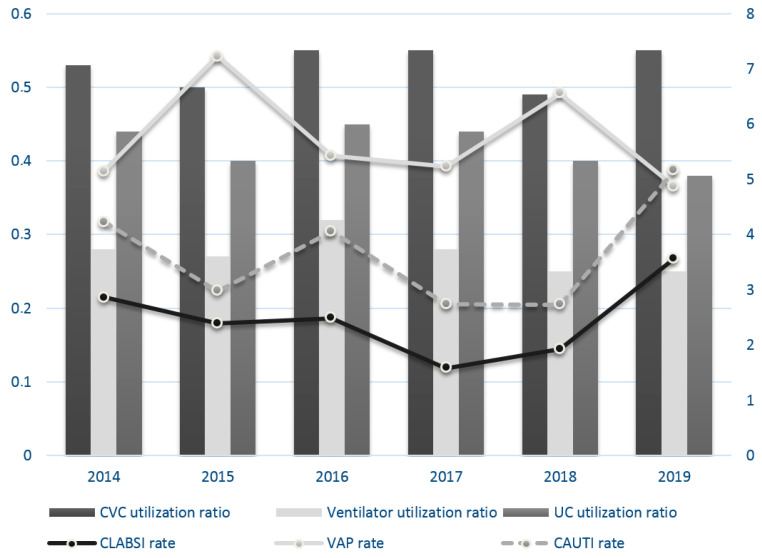
Device utilization ratio and device-associated infections. CAUTIs: catheter-associated urinary tract infections; CLABSIs: central-line-associated blood stream infections; CVC: central venous catheter; UC: urinary catheter; VAP: ventilator-associated pneumonia.

**Figure 2 children-09-01669-f002:**
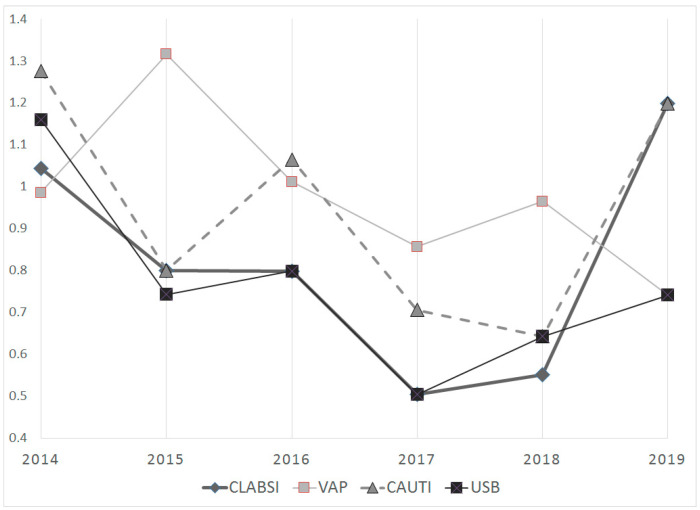
The cumulative incidence for each healthcare-associated infection per year. CAUTIs: catheter-associated urinary tract infections; CLABSIs: central-line-associated blood stream infections; USB: unknown-source bacteremia; VAP: ventilator-associated pneumonia.

**Figure 3 children-09-01669-f003:**
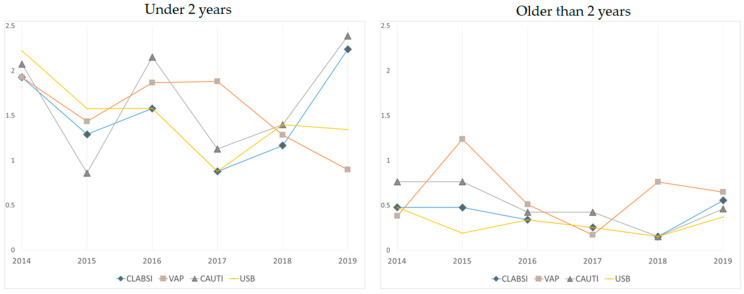
The cumulative incidence for each healthcare-associated infection per year, according to age. CAUTIs: catheter-associated urinary tract infections; CLABSIs: central-line-associated blood stream infections; USB: unknown-source bacteremia; VAP: ventilator-associated pneumonia.

**Figure 4 children-09-01669-f004:**
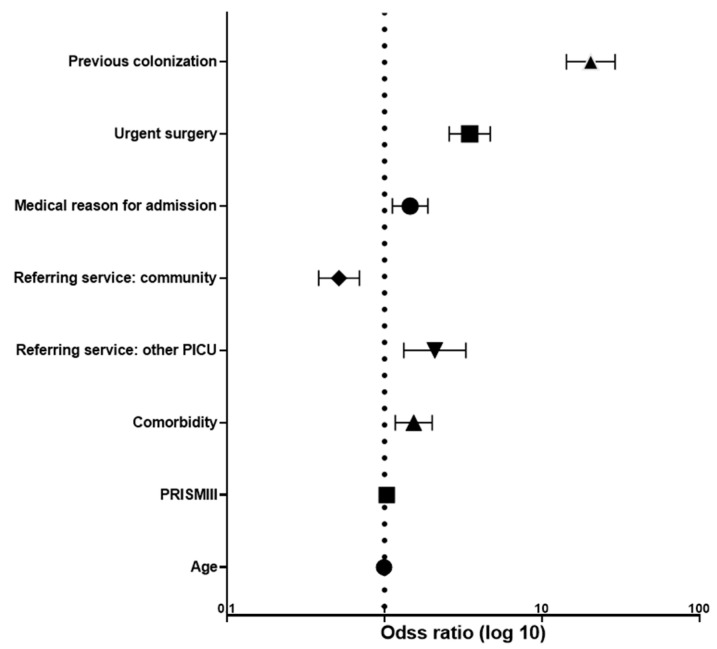
Forest plot representing the independent intrinsic risk factors for healthcare-associated infections. PICU: pediatric intensive care unit. PRISM III: Pediatric Risk of Mortality score.

**Figure 5 children-09-01669-f005:**
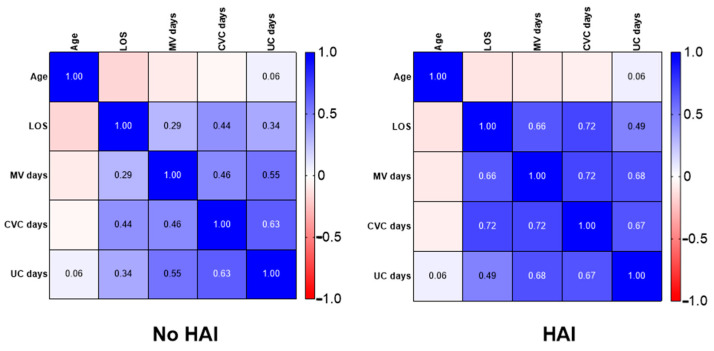
Correlation analysis for device exposure.

**Table 1 children-09-01669-t001:** Definitions of healthcare-associated infections according to ENVIN-Ped and based on CDC definitions [10,11].

Type of Infection	Definition
Central-line-associated bloodstream infection (CLABSI)	Primary blood stream infection (no other apparent source of infection) and positive blood cultures, all involving the same microorganism, fulfilling one of the following criteria:
	(a) Quantitative central venous catheter (CVC) culture ≥10^3^ CFU/mL; (b) Quantitative blood culture ratio of CVC blood sample/peripheral blood sample >5; (c) Differential delay in positivity of blood cultures: CVC blood sample culture positive two hours or more before peripheral blood culture; (d) Positive culture with the same microorganism found in pus at insertion site.
Ventilator-associated pneumonia (VAP)	**A. Clinical diagnosis:** (a) Presence of a new and persistent pulmonary infiltrate on one chest X-ray or CT scan in a previously healthy patient; OR (b) Two or more images suggestive of pneumonia in patients with underlying heart or lung disease; *AND at least one of the following:* - Fever ≥ 38 °C with no other discernable cause; - Leukopenia (<4000 WBC/mm^3^) or leukocytosis (≥12,000 WBC/mm^3^); AND at least one of the following (or at least two, if clinical pneumonia only): - Increased respiratory secretions, change in previous characteristics of sputum or sputum with purulent appearance; - New onset of cough, dyspnea and/or tachypnea; - Abnormal lung sounds, such as crackles, bronchial breath or wheezing; - Increased oxygen requirements or ventilatory demand;
	AND, depending on the diagnostic method used:
	**B. Bacteriological diagnosis:** (PN1). Positive quantitative culture from a minimally contaminated specimen: (a) Bronchoalveolar lavage (BAL) with a threshold of ≥1 × 10^4^ colony-forming units (CFUs)/mL or ≥5% of BAL-obtained cells containing intracellular bacteria upon direct microscope exam; (b) Protected specimen brush or protected distal aspirate, with a threshold of ≥1 × 10^3^ CFUs/mL. (PN2). Positive quantitative culture from a possibly contaminated specimen: (a) Quantitative culture from an endotracheal aspirate with a threshold of ≥1 × 10^6^ CFUs/mL. (PN3). Alternative microbiological methods: (a) Positive blood culture not related to another source of infection; (b) Positive growth in pleural fluid culture; (c) Pleural or pulmonary abscess, with positive needle aspiration; (d) Histological evidence of pneumonia; (e) Positive detection of viral antigen or antibodies in respiratory secretions; (f) Seroconversion; (g) Detection of viral antigen in urine. (PN4). Positive sputum culture or non-quantitative specimen culture. (PN5). No positive microbiology.
Catheter-associated urinary tract infection (CAUTI)	Defined in a patient who has at least one of the following symptoms, with no other recognized cause: (a) Fever > 38 °C, increased urgency and/or frequency, dysuria, or suprapubic tenderness; (b) Pyuria in urine specimen, with ≥10 WBC/mL or ≥3 WBC/high-power field of unspun urine;
	AND positive urine culture with a threshold of ≥1 × 10^5^ CFUs/mL with no more than two species of microorganisms in a patient that is not receiving antibiotic treatment; a threshold of <1 × 10^5^ CFUs/mL of one single microorganism in patients receiving antibiotic treatment.

**Table 2 children-09-01669-t002:** Description of the sample.

	Global *n* = 11,260	2014 *n* = 1724	2015 *n* = 1748	2016 *n* = 1877	2017 *n* = 1983	2018 *n* = 2176	2019 *n* = 1752	2014–2016 *n* = 5349	2017–2019 *n* = 5911	*p*
**Age (months)**	43 (10–115)	42.3 (9.6–105)	40.0 (8–111)	47.7 (10.9–118)	42.0 (9–115)	42.0 (10–121)	46.0 (11–121)	43 (10–111)	43 (10–119)	0.096
**Gender (male)**	6368 (56.6%)	971 (56.3%)	1019 (58.3%)	1057 (56.3%)	1121 (56.5%)	1242 (57.1%)	958 (54.7%)	3047 (57%)	3321 (56.2%)	0.404
**Referring service**										
Community	4119 (36.6%)	600 (34.8%)	648 (37.1%)	657 (35.0%)	799 (40.3%)	801 (36.8%)	614 (35.0%)	1905 (35.6%)	2214 (37.5%)	0.043
Pediatric hospitalization	6873 (61%)	1078 (62.5%)	1064 (60.9%)	1180 (62.9%)	1143 (57.6%)	1321 (60.7%)	1087 (62.0%)	3322 (62.1%)	3551 (60.1%)	0.027
Other PICU hospitalizations	261 (2.3%)	46 (2.7%)	33 (1.9%)	39 (2.1%)	39 (2.0%)	53 (2.4%)	51 (2.9%)	118 (2.2%)	143 (2.4%)	0.453
**Reason for admission**										
Medical	5469 (48.6%)	817 (47.4%)	839 (48.0%)	948 (50.5%)	901 (45.4%)	1045 (48.0%)	919 (52.5%)	2604 (48.7%)	2865 (48.5%)	0.821
Elective surgery	4338 (38.5%)	645 (37.4%)	657 (37.6%)	734 (39.1%)	816 (41.1%)	852 (39.2%)	634 (36.2%)	2036 (38.1%)	2302 (38.9%)	0.337
Urgent surgery	849 (7.5%)	158 (9.2%)	157 (9.0%)	119 (6.3%)	147 (7.4%)	149 (6.8%)	119 (6.8%)	434 (8.1%)	415 (7%)	0.028
Traumatic	595 (5.3%)	98 (5.7%)	93 (5.3%)	75 (4.0%)	119 (6.0%)	130 (6.0%)	80 (4.6%)	266 (5%)	329 (5.6%)	0.160
**Comorbidities**	2317 (20.6%)	396 (22.9%)	380 (21.7%)	377 (20.1%)	419 (21.1%)	388 (17.8%)	357 (20.4%)	1153 (21.6%)	1164 (19.7%)	0.015
Diabetes	108 (1%)	15 (0.9%)	19 (1.1%)	13 (0.7%)	17 (0.9%)	29 (1.3%)	15 (0.9%)	47 (0.9%)	61 (1%)	0.405
Kidney failure	343 (3%)	73 (4.2%)	56 (3.2%)	43 (2.3%)	62 (3.1%)	59 (2.7%)	50 (2.9%)	172 (3.2%)	171 (2.9%)	0.32
Immunosuppression	685 (6.1%)	131 (7.6%)	97 (5.5%)	128 (6.8%)	123 (6.2%)	107 (4.9%)	99 (5.7%)	356 (6.7%)	329 (5.6%)	0.016
Neoplasia	919 (8.2%)	146 (8.5%)	122 (7.0%)	158 (8.4%)	174 (8.8%)	159 (7.3%)	160 (9.1%)	426 (8%)	493 (8.3%)	0.466
Cirrhosis	97 (0.9%)	18 (1%)	22 (1.3%)	18 (1.0%)	13 (0.7%)	13 (0.6%)	13 (0.7%)	58 (1.1%)	39 (0.7%)	0.015
COPD	113 (1%)	20 (1.2%)	20 (1.1%)	19 (1.0%)	16 (0.8%)	25 (1.1%)	13 (0.7%)	59 (1.1%)	54 (0.9%)	0.314
Malnutrition	992 (8.8%)	167 (9.7%)	210 (12.0%)	182 (9.7%)	172 (8.7%)	130 (6.0%)	131 (7.5%)	559 (10.5%)	433 (7.3%)	<0.001
Transplantation	148 (1.3%)	23 (1.3%)	11 (0.6%)	28 (1.5%)	29 (1.5%)	35 (1.6%)	22 (1.3%)	62 (1.2%)	86 (1.5%)	0.169
**Microorganism colonization**	180 (1.6%)	33 (1.9%)	27 (1.5%)	23 (1.2%)	37 (1.9%)	34 (1.6%)	26 (1.5%)	83 (1.6%)	97 (1.6%)	0.706
MDRGN bacteria	39 (0.3%)	7 (0.4%)	4 (0.2%)	8 (0.4%)	9 (0.5%)	8 (0.4%)	3 (0.2%)	19 (0.4%)	20 (0.3%)	0.834
ESBL bacteria	77 (0.7%)	14 (0.8%)	4 (0.4%)	11 (0.6%)	22 (1.1%)	16 (0.7%)	10 (0.6%)	29 (0.5%)	48 (0.8%)	<0.001
Pseudomona spp.	19 (0.2%)	3 (0.2%)	7 (0.4%)	0 (0.0%)	0 (0.0%)	6 (0.3%)	3 (0.2%)	9 (0.2%)	10 (0.2%)	0.345
MRSA	17 (0.2%)	1 (0.1%)	3 (0.2%)	3 (0.2%)	5 (0.3%)	2 (0.1%)	3 (0.2%)	7 (0.1%)	10 (0.2%)	0.408

COPD: chronic obstructive pulmonary disease; ESBLs: extended-spectrum beta-lactamases; MRSA: methicillin-resistant *S. aureus*; MDRGN: multidrug-resistant Gram-negative; PICU: pediatric intensive care unit.

**Table 3 children-09-01669-t003:** Description of the healthcare-associated infection risk factors and mortality rate.

	Global *n* = 11,260	2014 *n* = 1724	2015 *n* = 1748	2016 *n* = 1877	2017 *n* = 1983	2018 *n* = 2176	2019 *n* = 1752	2014–2016 *n* = 5349	2017–2019 *n* = 5911	*p*
**HAI risk factors (1 or more)**	9558 (84.9%)	1507 (87.4%)	1512 (86.5%)	1586 (84.5%)	1695 (85.5%)	1801 (82.8%)	1457 (83.2%)	4605 (86.1%)	4953 (83.8%)	0.001
Antibiotics prior to PICU admission	2322 (20.7%)	303 (17.6%)	309 (17.7%)	394 (21.0%)	415 (20.9%)	441 (20.3%)	470 (26.8%)	1006 (18.8%)	1326 (22.4%)	<0.001
Antibiotics during PICU stay	8354 (74.2%)	359 (20.8%)	410 (23.5%)	513 (27.3%)	513 (25.9%)	634 (29.1%)	477 (27.2%)	4067 (76%)	4287 (72.5%)	<0.001
Previous surgery	4547 (40.4%)	682 (39.6%)	753 (43.1%)	654 (34.8%)	855 (43.1%)	910 (41.8%)	693 (39.6%)	2089 (39.1%)	2458 (41.6%)	0.006
Urgent surgery (during PICU stay)	904 (8%)	147 (8.5%)	177 (10.1%)	135 (7.2%)	178 (9.0%)	154 (7.1%)	113 (6.4%)	459 (8.6%)	445 (7.5%)	0.040
Central venous catheter	4858 (43.1%)	811 (47.0%)	758 (43.4%)	782 (41.7%)	868 (43.8%)	867 (39.8%)	772 (44.1%)	2351 (44%)	2507 (42.4%)	0.099
Mechanical ventilation	3788 (33.6%)	648 (37.6%)	654 (37.4%)	589 (31.4%)	743 (37.5%)	683 (31.4%)	471 (26.9%)	1891 (35.4%)	1897 (32.1%)	<0.001
Urinary catheter	5712 (50.7%)	924 (53.6%)	886 (50.7%)	946 (50.4%)	1059 (53.4%)	1037 (47.7%)	860 (49.1%)	2756 (51.5%)	2956 (50%)	0.108
External ventricular shunt	265 (2.4%)	64 (3.7%)	44 (2.5%)	39 (2.1%)	45 (2.3%)	37 (1.7%)	36 (2.1%)	147 (2.7%)	118 (2%)	0.009
Renal replacement therapy	218 (1.9%)	41 (2.4%)	35 (2.0%)	35 (1.9%)	40 (2.0%)	40 (1.8%)	27 (1.5%)	111 (2.1%)	107 (1.8%)	0.308
Parenteral nutrition	741 (6.6%)	154 (8.9%)	147 (8.4%)	114 (6.1%)	138 (7.0%)	105 (4.8%)	83 (4.7%)	415 (7.8%)	326 (5.5%)	<0.001
Neutropenia	324 (2.9%)	72 (4.2%)	53 (3.0%)	61 (3.2%)	52 (2.6%)	44 (2.0%)	42 (2.4%)	186 (3.5%)	138 (2.3%)	<0.001
ECMO	66 (0.6%)	14 (0.8%)	7 (0.4%)	14 (0.7%)	8 (0.4%)	10 (0.5%)	13 (0.7%)	35 (0.7%)	31 (0.5%)	0.367
**Mortality**	213 (1.9%)	44 (2.6%)	35 (2.0%)	45 (2.4%)	35 (1.8%)	30 (1.4%)	24 (1.4%)	124 (2.3%)	89 (1.5%)	0.002

ECMO: extracorporeal membrane oxygenation; HAIs: healthcare-associated infections; PICU: pediatric intensive care unit.

**Table 4 children-09-01669-t004:** Exposure data, healthcare-associated infection rates and device utilization ratio.

	Global	2014	2015	2016	2017	2018	2019	2014–2016	2017–2019	*p*
**Hospitals (*n*)**	33	27	27	25	24	29	26	26	26	-
**Total admissions (*n*)**	11,260	1724	1748	1877	1983	2176	1752	5349	5911	-
**Stays (days)**	69,512	11,743	11,635	10,972	11,556	12,880	10,726	34,350	35,162	-
**LOS (mean days, SD)**	6.17 (8.59)	6.81 (8.82)	6.66 (9.08)	5.85 (7.64)	5.83 (8.14)	5.92 (8.84)	6.12 (8.96)	6.42 (8.52)	5.95 (8.65)	0.004
**LOS (median days, IQR)**	3 (6–2)	4 (3–7)	4 (3–6)	3 (2–6)	3 (2–6)	3 (2–5)	3 (2–6)	4 (2–6)	3 (2–6)	<0.001
**HAI rate *n* (%)**	317 (2.8)	65 (3.8)	53 (3.0)	49 (2.6)	43 (2.2)	53 (2.4)	54 (3.1)	167 (3.1)	150 (2.5)	0.061
**Episodes of HAI *n* (%)**	390 (3.5)	77 (4.5)	64 (3.7)	69 (3.7)	51 (2.6)	61 (2.8)	68 (3.9)	210 (3.9)	180 (3.1)	0.011
**HAI/1000 patient days (‰)**	6.3	7.3	5.9	6.7	5.1	5.4	7.5	6.6	6	-
**CVC exposure data**	**Global**	**2014**	**2015**	**2016**	**2017**	**2018**	**2019**	**2014–2016**	**2017–2019**	** *p* **
CVC days	36,619	6275	5848	6035	6332	6246	5883	18,158	18,461	-
CLABSI episodes	90	18	14	15	10	12	21	47	43	0.368
CLABSI rate/1000 CVC days	2.46	2.87	2.39	2.49	1.58	1.92	3.57	2.59	2.33	-
CVC utilization ratio	0.53	0.53	0.50	0.55	0.55	0.49	0.55	0.53	0.53	-
**MV exposure data**	**Global**	**2014**	**2015**	**2016**	**2017**	**2018**	**2019**	**2014–2016**	**2017–2019**	** *p* **
MV days	19,115	3309	3181	3500	3251	3206	2668	9990	9125	-
VAP episodes	110	17	23	19	17	21	13	59	51	0.196
VAP rate/1000 MV days	5.75	5.14	7.23	5.43	5.23	6.55	4.87	5.91	5.59	-
Ventilator utilization ratio	0.27	0.28	0.27	0.32	0.28	0.25	0.25	0.29	0.26	-
**UC exposure data**	**Global**	**2014**	**2015**	**2016**	**2017**	**2018**	**2019**	**2014–2016**	**2017–2019**	** *p* **
UC days	29,144	5188	4703	4931	5120	5133	4069	14,822	14,322	-
CAUTI episodes	105	22	14	20	14	14	21	56	49	0.229
CAUTI rate/1000 UC days	3.60	4.24	2.98	4.06	2.73	2.73	5.16	3.78	3.42	-
UC utilization ratio	0.42	0.44	0.40	0.45	0.44	0.40	0.38	0.43	0.41	-

CAUTIs: catheter-associated urinary tract infections; CLABSIs: central-line-associated blood stream infections; CVC: central venous catheter; HAIs: healthcare-associated infections; LOS: length of stay; MV: mechanical ventilation: UC: urinary catheter; VAP: ventilator-associated pneumonia.

**Table 5 children-09-01669-t005:** Univariate analysis of risk factors for acquiring healthcare-associated infections.

	Global *n* = 11,260	No infection *n* = 10,943	Any HAI *n* = 317	*p*
**Age (months), median (IQR)**	43 (10–115)	45 (11–116)	8 (3–42)	<0.001
**Gender (male)**	6368 (56.6%)	6209 (56.7%)	159 (50.2%)	0.020
**PRISM III (points), median (IQR)**	2 (0–5)	2 (0–5)	7 (3–12)	<0.001
**Referring service**				
Community	4119 (36.6%)	4049 (37%)	70 (22.1%)	<0.001
Pediatric hospitalization	6873 (61%)	6665 (60.9%)	208 (65.6%)	0.090
Other PICU hospitalizations	261 (2.3%)	222 (2%)	39 (12.3%)	<0.001
**Admission cause**				
Medical	5469 (48.6%)	5292 (48.4%)	177 (55.8%)	0.009
Elective surgery	4338 (38.5%)	4232 (38.7%)	106 (33.4%)	0.059
Urgent surgery	849 (7.5%)	829 (7.6%)	20 (6.3%)	0.400
Traumatic	595 (5.3%)	581 (5.3%)	14 (4.4%)	0.484
**Comorbidities**	2317 (20.6%)	2206 (20.2%)	111 (35%)	<0.001
Diabetes	108 (1%)	107 (1%)	1 (0.3%)	0.376
Kidney failure	343 (3%)	313 (2.9%)	30 (9.5%)	<0.001
Immunosuppression	685 (6.1%)	648 (5.9%)	37 (11.7%)	<0.001
Neoplasia	919 (8.2%)	893 (8.2%)	26 (8.2%)	0.979
Cirrhosis	97 (0.9%)	90 (0.8%)	7 (2.2%)	0.008
COPD	113 (1%)	104 (1%)	9 (2.8%)	0.001
Malnutrition	992 (8.8%)	914 (8.4%)	78 (24.6%)	<0.001
Transplantation	148 (1.3%)	145 (1.3%)	3 (0.9%)	0.801
Microorganism colonization	180 (1.6%)	97 (0.9%)	83 (26.2%)	<0.001
MDRGN bacteria	39 (0.3%)	16 (0.1%)	23 (7.3%)	<0.001
ESBL bacteria	77 (0.7%)	37 (0.3%)	40 (12.6%)	<0.001
Pseudomona spp.	19 (0.2%)	12 (0.1%)	7 (2.2%)	<0.001
MRSA	17 (0.2%)	11 (0.1%)	6 (1.9%)	<0.001
*HAI risk factors* (≥1*)*	9558 (84.9%)	9241 (84.4%)	317 (100%)	<0.001
Antibiotics prior to PICU admission	2322 (20.7%)	2240 (20.5%)	92 (29%)	<0.001
Antibiotics during PICU stay	8354 (74.2%)	8044 (73.5%)	310 (97.8%)	<0.001
Previous surgery	4547 (40.4%)	4427 (40.5%)	120 (37.9%)	0.352
Urgent surgery (during PICU stay)	904 (8%)	812 (7.4%)	92 (29%)	<0.001
Central venous catheter	4858 (43.1%)	4555 (41.6%)	303 (95.6%)	<0.001
Mechanical ventilation	3788 (33.6%)	3502 (32%)	286 (90.2%)	<0.001
Urinary catheter	5712 (50.7%)	5417 (49.5%)	295 (93.1%)	<0.001
External ventricular shunt	265 (2.4%)	247 (2.3%)	18 (5.7%)	<0.001
Dialysis	218 (1.9%)	190 (1.7%)	28 (8.8%)	<0.001
Parenteral nutrition	741 (6.6%)	631 (5.8%)	110 (34.7%)	<0.001
ECMO	66 (0.6%)	43 (0.4%)	23 (7.3%)	<0.001

COPD: chronic obstructive pulmonary disease; ECMO: extracorporeal membrane oxygenation; ESBLs: extended-spectrum beta-lactamases; HAIs: healthcare-associated infections; MRSA: methicillin-resistant S. aureus; MDRGN: multidrug-resistant Gram-negative; PICU: pediatric intensive care unit; PRISM III: Pediatric Risk of Mortality score.

## Data Availability

Not applicable.

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
