# Peer review of "Device Exposure and Patient Risk Factors’ Impact on the Healthcare-Associated Infection Rates in PICUs"

_children, 2022, doi:10.3390/children9111669_

Round 1
Reviewer 1 Report
The manuscript entitled “Device exposition and patient risk factors impact in the healthcare-associated infection rate in PICUs” has been reviewed. The authors try to the risk factors of patients in PICU about DA-HAI. They also want to demonstrate the effects of HAI Zero Bundles. From the results and discussion, the bundles appeared to reduce the HAI rates. However, some questions about the manuscript require further explanation.
1. It appeared to be several different Zero Bundles implemented in the same time period. It would be better to provide a brief introduction to the bundles implemented in PICU. Was it possible that more than one bundles were applied at the same patient in the same time? How were these data interpreted? Could it be the effect of one bundle instead of several bundles?
2. In the result section, the format of Table 1 and Table 2 required further adjustment. They are quite difficult to read. I suggest authors further divide Table 2 into more smaller tables according different category.
3. The age of the patients varied from 1 month to 18 years. I would suggest the patients should be divided into different age groups, and re-analyzed the effects of Zero Bundles.
4. Besides the age, severity of the patients while admitted to PICU might also affect the outcome of bundle care, such as the administration of device, the duration of device usage, and the duration of staying in PICU. If it is possible, analyzed the data according to the severity (such as PRISM III score) should be performed in different score groups.
5. It is understandable that the author wanted to show the overall effect of Zero Bundles on DA-HAI rate. I still suggest that it would be more suitable to discuss why some bundles were more or less effective.
Reviewer 2 Report
General Comment:
The topic is not new, but there are not many studies in the literature about the “Zero Bundle” effect.
The paper is well structured.
Title: Title is ok, I suggest to add “Spain”.
Abstract: Please explain the abbreviations like PICUs – CLABSI – VAP – CAUTI . They are defined in the main text, but you must also express them here.
Keyword: I suggest to add the keyword “SPAIN” , and if possible to delete “bacterial colonization”
Introduction: This part is well written, but here you can explain better the “Zero Bundles” sets and you can focus better on what has already been studied in the literature about the Healthcare-associated infections.
Material and Methods: This section is correctly divided into sub-headings, which makes it easier to read. The chosen period of the analysis is only 3 months per year: Have you verified if this aspect can introduce a BIAS?
Another problem to take into account is that the different PICUs implemented the HAI zero bundles in different steps.
Line 9: Authors say about a survey to determine when the PICUs implement the HAI zero bundles, but they did not explain anything about it. What kind of survey? Which questions?
Table 1 : Reference n.12 it ‘ s not so understandable. The table formatting must be revised. It is not properly divided into sections and the division into the three “bundles” is difficult to see visually.
I recommend using a flowchart to summarise all the steps in this section.
Results: I suggest to put the text of this section before Table 2 and 3.
Table 2: need an overhaul in formatting as I would suggest decreasing the text's size and having all data in one row of the table for alignment when possible. Results are a little bit confusing. On page 8 when the table continues, you have to add the row again with the divisions by year for better reading.
The figures in this section (1-2-3-4) are fine.
Discussion: In this section, the references are updated. One of the most significant limitations to this study is the geographical location restricted to only a part of the PICUs in Spain. A Europe-wide analysis would be desirable.
In addition, a section about the possible limits of the paper should be presented.
Reviewer 3 Report
The manuscript is very important in the development of the vulnerable pediatric patiens treament and life expectancy in intensive caare units. The manuscript has a very good starting point for an international research sequence for faighting againsts the pediatric patients Healthcare-associated infections related to device use.
The manuscript has both human related and financial meaning for the health care.
